# The Lack of the Association of the CCR5 Genotype with the Clinical Presentation and Frequency of Tick-Borne Encephalitis in the Polish Population

**DOI:** 10.3390/pathogens11030318

**Published:** 2022-03-04

**Authors:** Sambor Grygorczuk, Justyna Dunaj-Małyszko, Artur Sulik, Kacper Toczyłowski, Piotr Czupryna, Agnieszka Żebrowska, Miłosz Parczewski

**Affiliations:** 1Department of the Infectious Diseases and Neuroinfections, Faculty of Medicine, Medical University of Białystok, ul. Żurawia 14, 15-540 Białystok, Poland; justyna.dunaj@umb.edu.pl (J.D.-M.); piotr.czupryna@umb.edu.pl (P.C.); 2Department of the Pediatric Infectious Diseases, Faculty of Medicine, Medical University of Białystok, ul. Jerzego Waszyngtona 17, 15-274 Białystok, Poland; artur.sulik@umb.edu.pl (A.S.); kacper.toczylowski@umb.edu.pl (K.T.); 3Regional Centre of Transfusion Medicine, ul. Marii Skłodowskiej-Curie 23, 15-950 Białystok, Poland; azebrowska@rckik.bialystok.pl; 4Department of the Infectious Diseases, Tropical Diseases and Acquired Immunodeficiencies, Pomeranian Medical University, ul. Arkońska 4, 71-455 Szczecin, Poland; mparczewski@yahoo.co.uk

**Keywords:** tick-borne encephalitis, chemokine receptor, CCR5Δ32, genetic association study

## Abstract

Background: The host factors influencing the susceptibility to and the severity of tick-borne encephalitis (TBE) are poorly defined. The loss-of-function *Δ32* mutation in the chemokine receptor gene *CCR5* was identified as a risk factor for West Nile encephalitis and possibly for TBE, suggesting a protective role of CCR5 in *Flavivirus* encephalitis. Methods: We studied the *CCR5* genotype in 205 TBE patients stratified by a clinical presentation and 257 controls from the same endemic area (Podlasie, Poland). The genotype distribution between the groups and differences between TBE patients with different genotypes were analyzed. Results: There were 36 (17.6%) *CCR5**Δ32* heterozygotes and 3 (1.5%) homozygotes in the TBE group, with no statistically significant difference in comparison with the controls. The *CCR5**Δ32* allele did not associate with the clinical presentation or the severity of TBE. The cerebrospinal fluid (CSF) inflammatory parameters did not differ between the wild-type (*wt/wt*) and *wt/**Δ32* genotype patients. The TBE clinical presentation and CSF parameters in three *Δ32/**Δ32* homozygotes were unremarkable. Conclusions: The lack of association of *CCR5**Δ32* with the risk and clinical presentation of TBE challenges the suspected CCR5 protective role. CCR5 is not indispensable for the effective immune response against the TBE virus.

## 1. Introduction

Tick-borne encephalitis virus (TBEV) is a *Flavivirus* transmitted by *Ixodes* ticks and endemic in a large part of the moderate climate zone of Eurasia. The majority of infections with the European TBEV subtype (EuTBEV) remain either asymptomatic [1,2] or too mild to draw medical attention [3] and are recognized only accidentally or by dedicated surveys. The clinically overt cases differ in severity, from uncomplicated meningitis to life-threatening encephalitis or encephalomyelitis [4,5]. Factors contributing to a variable outcome of TBEV infections are not well understood, but the individual variability of the host’s immune response is considered to play a decisive role [4,6]. Animal models suggest a multi-step process of the disease progression, which may be shaped by the host response at several levels: during the peripheral spread of TBEV, the invasion across the blood–brain barrier (BBB), or the resulting central nervous system (CNS) involvement [7,8]. Several polymorphisms in the genes involved in the inflammatory and antiviral response are associated with the risk and/or severity of TBE in exposed human populations, constituting potential risk factors and offering an insight into the disease pathogenesis [9,10,11,12].

Chemokines are a family of small cytokines that are chemotactic for leukocytes, among other functions involved in attracting different leukocyte populations into an inflammatory focus. The chemokines interact with their specific receptors, which are expressed selectively on different leukocyte populations in a regulated manner. The changeable pattern of the chemokine ligand and receptor expression allows for the differential control of the leukocyte migration and thus may determine the composition of the inflammatory infiltrate and the features of the local immune response [13,14]. In CNS infections, it is reflected by a composition of the leukocyte infiltrate in the brain parenchyma and the pleocytosis of the cerebrospinal fluid (CSF), the latter being available for a clinical examination and an important diagnostic criterion [14,15]. In TBE, the CSF cytosis is dominated by Th CD4+ lymphocytes, mostly of the Th1 subset, with the addition of Tc CD8+ cells [16]. The detailed protective and pathogenic effects exerted by these lymphocyte populations were evaluated mainly in post-mortem studies, animal models, and in a few clinical-setting studies, and, to a large extent, remain debatable [17,18,19,20,21].

CCR5 is a receptor for CCL3, CCL4, and CCL5 chemokines, expressed on T lymphocytes in a constitutional and induced manner [14,22]. It is also a human immunodeficiency virus (HIV) co-receptor targeted by a class of antiretroviral drugs, the entry inhibitors, represented by maraviroc [23,24,25]. In viral encephalitis, it may be involved in a lymphocyte migration into and within the CNS, microglia activation, and an intrathecal inflammatory response, contributing to disease control, but also potentially to an immune-mediated pathology [21,26]. In TBE, the activated intrathecal T CD4+ and T CD8+ lymphocyte populations are enriched in CCR5-positive cells [21]. Mice unable to express CCR5 have unfavorable alterations of the intrathecal response to several *Flavivirus* species, including the West Nile virus (WNV), the Japanese encephalitis virus, and the Langat virus, closely related to TBEV [27,28,29]. In humans, this condition may be mimicked in the bearers of the *Δ32* deletion in the *CCR5* gene, manifesting itself with a lack of a functional CCR5 in homozygotes and a several-fold reduced expression in heterozygotes [22,24,25]. The *CCR5**Δ32* homozygosity correlated with the increased risk of asymptomatic WNV infection and with its fatal outcome in the genetic association studies in North America [30,31], which raised concerns about increased susceptibility to WNV in patients receiving maraviroc [23,31]. Consequently, an association of *CCR5**Δ32* with the risk of symptomatic TBEV infection has been reported in the Lithuanian population, and a tendency for an association with the clinical severity of asymptomatic TBE was suggested, but not confirmed, in a follow-up study [9,10]. Some mechanistic explanations of these associations have been proposed but remain unproven. The protective effect of CCR5 could be exerted during the early peripheral phase of the *Flavivirus* infection, when it could reduce the risk of a symptomatic disease and neuroinvasion. A decreased CCR5 expression could also decrease Th lymphocyte migration from the periphery into the CNS, hampering the development of the protective intrathecal immune response and increasing the risk and severity of the neurologic involvement [27]. As the CSF of TBE patients contains a large fraction of CCR5-negative T lymphocytes [21,32], other pathways must supplement the chemotactic effect of CCR5 signaling and should be able to compensate for its dysfunction to some degree. Still, the paucity of a functional CCR5 could delay the lymphocyte influx and alter the balance between different intrathecal leukocyte subpopulations, resulting in a less effective and/or more immunopathogenic local response, as observed in some animal models of *Flavivirus* encephalitis [28,29]. However, the character and extent of these alterations in human TBE remain unknown.

In our previous studies, we were unable to confirm the association between the *CCR5* genotype and the risk and severity of TBE, and we have found that *CCR5 wt/**Δ32* heterozygotes were able to mount a normal intrathecal immune response to TBEV [32,33]. Currently, we have attempted to verify these findings in a larger study group and to evaluate the clinical presentation of TBE in patients heterozygous and homozygous for *CCR5**Δ32* mutation in more detail.

## 2. Results

To assess if the *CCR5* genotype influences the risk of symptomatic TBEV infection in exposed individuals, we have first compared the distribution of the genotypes between TBE patients and healthy controls inhabiting the same highly endemic area in Podlasie in the northeast of Poland. There were 39 bearers of the *Δ32* allele in the TBE group (19.1%), including 36 *wt/**Δ32* heterozygotes and three *Δ32/**Δ32* homozygotes, which did not differ from the frequency in the control group and gave no hint of any association of the *CCR5**Δ32* allele with TBE occurrence (Table 1, top two rows).

Next, we have compared the distribution of the CCR5 genotypes between the subgroups of TBE patients defined by qualitative and semi-quantitative clinical variables. Because there were only three *Δ32/**Δ32* homozygotes, in the formal analysis, they were pooled with *wt/**Δ32* heterozygotes, and the distributions of the *Δ32*-negative and the *Δ32*-positive genotypes were compared. The results are presented in Table 1. The *Δ32* allele did not associate significantly with the clinical presentation (defined as meningitis, meningoencephalitis, or meningoencephalomyelitis), the severity of the neurologic involvement (scored in a simplified four-grade scale from absent to severe), the presence and severity of consciousness abnormalities, and the presence of two frequent neurologic manifestations (paresis and cerebellar syndrome), as well as the history of a clinically distinct peripheral phase. Of note, there were some trends for an association of *Δ32*, particularly of the *wt/**Δ32* genotype, with a more severe TBE (a disease graded as severe, moderate to severe consciousness abnormalities, paresis, and monophasic presentation), but they all depended on small patient numbers and were well below the level of the statistical significance.

There was no significant difference in the basic CSF inflammatory parameters between the TBE patients with *wt/wt* and *wt/**Δ32* genotypes, although the latter tended to have somewhat higher CSF lymphocyte counts (Table 2).

As *CCR5* heterozygotes, who constituted the majority of *Δ32*-bearing patients, could potentially compensate for the impaired CCR5 expression by the upregulation of the CCL5 and other CCR5 ligands, the normal TBE presentations in them do not exclude the involvement of CCR5 signaling. We have used the opportunity for the detection of three TBE *Δ32* homozygotes, by definition incapable of any functional CCR5 expression, to further constrain its possible pathophysiological role. The main clinical and laboratory findings in these patients are presented in Table 3. 

In general, the clinical TBE presentation in *CCR5*
*Δ32/**Δ32* patients was unremarkable and relatively mild. None of them had severe disease, and two presented with meningitis with no neurologic complications. Their CSF parameters were also within the range of the values found in *wt/wt* homozygotes and individually variable, without any consistent trend common to all of them.

## 3. Discussion

Our results do not exclude the possibility of CCR5 being expressed and participating in the immune response in human TBE, as suggested by different lines of evidence revealed in previous studies [9,10,29,32]. However, they strongly suggest that (1) CCR5 is not essential in preventing symptomatic disease and CNS involvement in TBEV-exposed individuals, which means it probably does not play a decisive role in the primary infection focus or during the viremic phase; (2) it is not indispensable in controlling CNS infection by TBEV. Our results differ from the previous findings of Kindberg et al. and Mickiene et al. Kindberg et al. compared 129 TBE patients from Lithuania with 134 healthy controls and groups of historical population controls from Lithuania and Sweden, finding a significantly higher *CCR5**Δ32* allele frequency in the TBE group. The study identified three *CCR5**Δ32/**Δ32* homozygotes (2.3%) in the TBE cohort and none among TBEV-naive controls and found a trend for a higher frequency of *wt/**Δ32* genotype in TBE than in controls (22.5% and 16.4%, respectively). There was also a tendency for an association of the *Δ32* allele with a more severe TBE presentation [9]. The study by Mickienė et al. compared three cohorts of pediatric and adult TBE patients (349 TBE cases) with 135 healthy subjects. It corroborated the association of *CCR5**Δ32* with the risk of TBE but was unable to replicate the association with clinical severity [10]. Both studies included groups of patients with aseptic non-TBE meningitis/meningoencephalitis who presented with the *CCR5**Δ32* frequency that was not different from healthy controls and lower than in TBE patients. The results were suggestive of a specific role of CCR5 in TBE and not in other viral CNS infections [9,10]. In our previous study including patients and controls from the northeast area of Poland, we were unable to replicate these findings, but our study group was relatively small (59 TBE patients and 57 controls) [33]. Currently, however, we have obtained similar negative results with a much larger study population. The variability of TBEV strains and/or genetic background in study populations could possibly contribute to the difference between our current results and those of Kindberg et al. and Mickienė et al. despite the studies being conducted in adjacent geographic areas. Interestingly, similarly to us, Barkash et al. did not detect an association of the CCR5 genotype with the susceptibility to TBE in the Russian population in the area dominated by the Siberian TBEV subtype [34]. However, both in Lithuania and in northeast Poland, only the European (Western) TBEV subtype has been detected, which additionally shows relatively little variation between the strains isolated in different European sites [35,36,37]. All of this makes the intra-strain virus variability an unlikely explanation of the difference between the results obtained in the adjacent areas of Lithuania and Poland. As the discussed effects of the *CCR5* genotype on the TBE rate and presentation are rather subtle, the discrepancy can be overcome by further studies with a higher statistical power.

As we were able to document a normal clinical presentation and CSF cellular parameters in the *Δ32/**Δ32* homozygotes, we can be confident that CCR5 is not indispensable for a normal inflammatory response and lymphocyte migration into the CSF in TBE. This should reduce the concerns about the security of anti-CCR5 therapies in persons exposed to TBEV. Based on our current data, we cannot exclude a weak trend for a more severe presentation of TBE and, somewhat counterintuitively, higher lymphocytic pleocytosis in the *wt/**Δ32* heterozygotes. Such a tendency would be consistent with the findings of Kindberg et al. [9] and could be attributed to changes in the activity of other immune mediators in the face of a reduced CCR5 expression, as suggested by animal studies [28,29]. However, before any further interpretation attempts are made, the tentative trend needs to be verified in further studies.

In conclusion, we suggest that either the CCR5 axis is not essential in the response to TBEV in humans, and its reduced expression does not influence the course of the infection, or, more likely, it is engaged in that response, but not indispensable, and can be effectively replaced by other signaling routes. In the second case, some possibly unfavorable alterations of the immune response to the TBEV may be expected in persons with a hampered CCR5 expression, but any potential clinical effects would be limited and must be confirmed by follow-up research.

## 4. Materials and Methods

The study group consisted of 205 TBE patients hospitalized from 2016 to 2020 in the Department of the Infectious Diseases and Neuroinfections and the Department of the Pediatric Infectious Diseases of the Medical University in Białystok (Podlasie, Poland). All patients had an epidemiologic and clinical history consistent with the diagnosis of TBE, corroborated by either the detection of specific anti-TBEV IgM and/or IgG antibodies in serum and/or CSF on admission or by seroconversion, fulfilling the criteria of a confirmed TBE case [38]. The serologic testing was performed with FSME/TBE Elisa IgM and FSME/TBE Elisa IgG from VIROTECH Diagnostics GmbH (Germany). There were 134 males (65.4%) and 74 females (34.6%), from 6 to 88 years old (mean 45.7 years), including 12 pediatric patients (6–17 years old) and 193 adults. Three patients did not undergo lumbar puncture because of contraindications, and five had a borderline CSF cytosis (8–14 cells/μL); all the others (198; 96%) had a confirmed pleocytosis > 15 cells/μL. All but two patients (203; 99%) were IgM-positive towards TBEV either in serum or CSF and most of them both in serum and CSF simultaneously. The patients were not screened for a potential cross-reactivity with related viruses, but as there are no known *Flavivirus* species other than the TBEV circulation in the study area, we consider the probability of such infections to be extremely small. Most patients denied any previous anti-TBEV vaccination, but 2 (1.0%) reported having undergone an incomplete vaccination scheme—they were not excluded, as their both clinical and serologic presentation was consistent with a current infection with TBEV. The patients were stratified according to the clinical severity and main neurologic symptoms. Patients with meningeal syndrome but no focal neurologic deficits or altered consciousness were classified as having uncomplicated meningitis (M), whereas patients with an altered mental status and/or any focal CNS involvement were classified as having meningoencephalitis (ME) or meningoencephalomyelitis (MEM). The CNS abnormalities were graded as absent (corresponding to M); mild (paresthesia, pathologic reflexes, nystagmus, mild gait disorders); moderate (focal symptoms including paresis and/or lethargy); and severe (multiple or severe focal deficits, disorientation, loss of consciousness). Consciousness abnormalities were stratified as mild (lethargy) through moderate (agitation or disorientation) to severe (loss of consciousness), paresis, and cerebellar syndrome—as present or absent. The patients were also stratified as having either a classical biphasic course of TBE with a distinct peripheral phase before the onset of meningitis or a more rapidly progressive, monophasic presentation. The CSF inflammatory parameters (pleocytosis with leukocyte differential, protein, and albumin concentration) were evaluated by standard laboratory techniques in a hospital diagnostic laboratory.

The control group was recruited from a population of 300 healthy adult blood donors applying to the Regional Centre of Transfusion Medicine in Białystok, within the study area. Each blood donor was asked to fill out a short questionnaire asking about 1) a history of a diagnosed tick-borne encephalitis, or 2) a history of vaccination against TBE, and the participants reporting either of these were excluded, leaving a group of 257 controls, including 192 males (74.7%) and 65 females (25.3%), ranging from 18 to 63 years old (mean age of 34.8 years).

Because both the control and patient groups were recruited in the same geographic area and were ethnically uniform, we did not undertake further demographic stratification. The patients and controls gave written, informed consent for inclusion. The study was approved by the Ethics Committee of the Medical University in Białystok (approval no R-I-002/308/2019). 

A sample of 1 mL of venous blood was obtained in an EDTA-coated tube during the hospital stay in patients and directly before a blood donation in controls, kept in 4–5 °C for no more than 72 h, frozen to −20 °C, and stored till DNA extraction. QIAamp DNA Blood Mini Kit (QIAgen, Hilden, Germany) was used to extract genomic DNA, following the manufacturer’s protocol. DNA was re-suspended in 200 µL of AE buffer (QIAgen, Hilden, Germany) and stored at 4 °C for further analyses. The genotyping was performed in the laboratory of the Department of Infectious Diseases, Tropical Diseases, and Acquired Immunodeficiencies of Pomeranian Medical University in Szczecin. To analyze *CCR5Δ32* variation, PCR with sequence-specific primers was used as described previously [39]. Visualization under UV light was performed after electrophoresis on the 2.5% agarose gel (SIGMA, Saint Louis, MO, USA) stained with DNA-star dye (Lonza Inc., Rockland, ME, USA). Genotyping was successful in all 465 samples. No deviation from Hardy-Weinberg equilibrium WHE was detected.

We have used Pearson’s chi-square test to compare the wt and *Δ32* allele frequency in TBE patients and controls, as well as in the TBE patient subgroups defined by clinical presentation. We have compared basic CSF parameters (pleocytosis, lymphocyte count, neutrophil count, protein, and albumin concentration) between the patients with *wt/wt* and *wt/**Δ32* genotypes with the U Mann–Whitney test. *p* < 0.05 was considered significant. The analysis was separately performed with the exclusion of 8 patients with no confirmed pleocytosis, one of whom had *wt/**Δ32* and 7 *wt/wt* genotype, with a marginal influence on the results and no change to their statistical interpretation.

## Figures and Tables

**Table 1 pathogens-11-00318-t001:** Distribution of *CCR5* genotypes in the study cohort. Frequencies of *CCR5* genotypes and the *CCR5* allele prevalence in patients with tick-borne encephalitis (TBE) stratified according to clinical variables and in healthy controls from the same area. The frequencies of *wt/wt* and combined *wt/**Δ32* and *Δ32/**Δ32* genotypes, as well as *wt* and *Δ32* allele prevalence, did not differ significantly between the groups.

Group	*CCR5* Genotype Prevalence ^a^			CCR5 Allele Prevalence
	*wt/wt*	*wt/* *Δ32*	*Δ32/* *Δ32*	(*wt/**Δ32* Allele)
Healthy controls (*n* = 265)	212 (82.5%)	41 (16.0%)	4 (1.6%)	0.905/0.095
TBE (*n* = 205)	166 (81.0%)	36 (17.6%)	3 (1.5%)	0.898/0.102
Clinical presentation				
meningitis (M) (*n* = 109)	88 (80.7%)	19 (17.4%)	2 (1.8%)	0.894/0.106
meningoencephalitis (ME) (*n* = 77)	63 (81.8%)	13 (16.9%)	1 (1.3%)	0.903/0.097
meningoencephalomyelitis (MEM) (*n* = 19)	15 (78.9%)	4 (21.1%)	0 (0.0%)	0.895/0.105
Severity of ME/MEM				
mild (*n* = 43)	36 (83.7%)	6 (14.0%)	1 (2.3%)	0.907/0.093
moderate (*n* = 36)	30 (83.3%)	6 (16.7%)	0 (0.0%)	0.917/0.083
severe (*n* = 17)	12 (70.6%)	5 (29.4%)	0 (0.0%)	0.853/0.147
Consciousness abnormalities				
absent (*n* = 155)	128 (82.6%)	24 (15.4%)	3 (1.9%)	0.903/0.097
present (*n* = 50):	38 (76.0%)	12 (24.0%)	0 (0.0%)	0.880/0.120
mild (*n* = 33)	26 (78.8%)	7 (21.2%)	0 (0.0%)	0.894/0.106
moderate (*n* = 9)	6 (66.7%)	3 (33.3%)	0 (0.0%)	0.833/0.167
severe (*n* = 8)	6 (75.0%)	2 (25.0%)	0 (0.0%)	0.875/0.125
Paresis				
absent (*n* = 187)	152 (81.3%)	32 (17.1%)	3 (1.6%)	0.898/0.102
present (*n* = 18)	14 (77.8%)	4 (22.2%)	0 (0.0%)	0.889/0.111
Cerebellar syndrome				
absent (*n* = 166)	134 (80.7%)	30 (18.1%)	2 (1.2%)	0.898/0.102
present (*n* = 39)	32 (82.1%)	6 (15.4%)	1 (2.6%)	0.897/0.103
Disease course				
monophasic (*n* = 110)	87 (79.1%)	21 (19.1%)	2 (1.8%)	0.886/0.114
biphasic (*n* = 92)	78 (84.8%)	13 (14.1%)	1 (1.1%)	0.918/0.082

^a^ number of cases (frequency of a genotype in %); *wt*—wild type.

**Table 2 pathogens-11-00318-t002:** The cerebrospinal fluid (CSF) parameters in TBE patients stratified according to CCR5 genotype. The median values of the basic CSF parameters on admission in all TBE patients and in subgroups with CCR5 *wt*/*wt* and *wt*/Δ32 genotype. There was no significant difference between the compared genotypes.

CCR5 Genotype	All	*wt/wt*	*wt/* *Δ* *32*
pleocytosis ^a^	92	91	103
lymphocyte count ^a^	58	57	74
protein ^b^	0.65	0.66	0.63
albumin ^b^	0.447	0.447	0.444

^a^—cells/μL; ^b^—g/L; *wt*—wild type.

**Table 3 pathogens-11-00318-t003:** The clinical and laboratory characteristics of the individual *CCR5*
*Δ32/**Δ32* TBE patients. The selected demographic, clinical, and laboratory parameters in three identified TBE patients with the *CCR5*
*Δ32/**Δ32* genotype.

No	Sex	Age	Presentation	Altered Consciousness	Neurological Symptoms	CSF Parameters on Admission			
						Pleocytosis ^a^	Lymphocytes ^a^	Protein ^b^	Albumin ^b^
1	m	34	M	no	no	58	NA	0.35	0.239
2	m	46	M	no	no	63	45	1.12	0.787
3	f	38	ME	no	cerebellar syndrome	193	58	0.76	0.508

^a^—cells/μL; ^b^—g/L; m—male; f—female; M—meningitis; ME—meningoencephalitis; NA—non-available.

## Data Availability

The data presented in this study are available on request from the corresponding author.

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
