# Peer review of "The Lack of the Association of the CCR5 Genotype with the Clinical Presentation and Frequency of Tick-Borne Encephalitis in the Polish Population"

_pathogens, 2022, doi:10.3390/pathogens11030318_

Round 1

Reviewer 1 Report

Dear authors,

Congratulations for this fine study. I have no major comments, but I would like to mention few things:

-Please specify what analysis was used for detection of anti-TBEV Abs, have you considered vaccination status in patients who had only serum analysis? Additionally, have you considered cross-reactivity of other anti-flavivirus antibodies with TBEV antigens?

-Control group: Although I approve your selection of control subjects, I would link for you to consider possible presence of bias. More precisely, would you consider that more suitable comparing group would be the persons who have been exposed to TBEV (IgG seroreactive) bud did not develop TBE? In this case you have all subjects who have actually been exposed to virus, in current control group situation in heterogenous.

Discussion:

You mentioned in line 186 TBEV-strain dependent variation. Would you please elaborate possible influence of CCR5 genotype on infections with different TBEV strains (-Eu, -Sib, -FE). I think that this is interesting especially because several TBEV strains are present in this part of Europe. 

Author Response

Dear Sir/Madam

Thank you very much for your comments on the paper.

Regarding your suggestions:

  • The patients were tested for anti-TBEV antibodies with FSME/TBE Elisa IgM and IgG kits from Virotech.
  • The patients with a TBE suspicion were routinely asked about having received an anti-TBE vaccine. Two reported a history of an uncomplete anti-TBEV vaccination: one had high a anti-TBEV IgM and IgG titer in CSF, the other had only specific IgG detected in CSF, but was IgM-positive in serum.
  • There are no other artropod-borne Flavivirus species circulating in the study area. Patients were routinely asked about travel history so imported cases of other Flavivirus infections should have been identified. IgG cross-reactivity because of past infections when travelling abroad cannot be excluded with absolute certainity. However, it was not suspected on clinical or epidemiologic grounds in any of the patients. Moreover, in a vast majority of patients the diagnosis was supported by a specific IgM detection in CSF, so we believe any false diagnoses because of the cross-reactivity are highly unlikely.
  • We have added some information to clarify the above issues at the beginning of the Matherials and methods section at the bottom of p. 6.
  • We agree that persons with a serologic proof of an asymptomatic TBEV infection would make ideal controls, however, it would need a prohibitively large screening campaign to select such a control group of the relevant size. The methodology for a control group selection in our study was comparable with studies on CCR5 genotype in WNV and TBE we quote and compare our results to.
  • The published data we are aware of are scarce, but we have added further discussion on the possibility of TBEV subtype/strain influence on the results, with 4 additional references (34-37).

Reviewer 2 Report

The authors describe a for years already ongoing study on the importance of chemokine recpetors and interleukins for the severeness of TBE overt disease.

The results are therefore interesting although not extraordinary. The paper is written in good style, all results are presented well, the methods are sound. The statistics should be reviewed by a experieinced statistician.

Otherwise I do not have any comments.

Author Response

Dear Sir/Madam,

Thank you very much for your comments.

As the statistical methodology applied was relatively simple and straightforward, the analysis was carried out by one of the authors using a commercial software package. There were no apparent differences between the groups on the initial data inspection, which agrees excellently with the negative result of the formal statistical analysis, and which has given us credence in the validity of our conclusions.

If requested, we could provide the raw data for an independent check and/or proceed with re-analysis.

Your sincerely

Sambor Grygorczuk

Reviewer 3 Report

The manuscript "The lack of the association of the CCR5 genotype with the clinical presentation and frequency of tick-borne encephalitis in Polish population" represents the extension of the previous research of authors. The main aim was to verify in the larger study group the previously observed lack of association between CCR5 genotype and the risk and severity of TBE. The findings of this study are both fundamental, to clarify the factors contributing to the variable outcome of TBEV infection, and practical - to estimate the safety of therapy with anti-CCR5 in the population of people exposed to TBEV.  

The overall quality of the manusript is high, it's written excellently, with clearly presented results, and discussed accordingly. 

Author Response

Dear Sir/Madam,

Than you very much for your comments.

Your sincerely

Sambor Grygorczuk

Round 2

Reviewer 1 Report

Dear authors,

Thank you for your response. I don't have any additional questions or comments.